

# Climate and Ocean Circulation in the Aftermath of a Marinoan Snowball Earth

Lennart Ramme[1,2] and Jochem Marotzke[1,3]

[1]Max-Planck-Institute for Meteorology, Hamburg, Germany
[2]International Max Planck Research School on Earth System Modelling, Hamburg, Germany
[3]Center for Earth System Research and Sustainability (CEN), Universität Hamburg, Germany

**Correspondence:** Lennart Ramme (lennart.ramme@mpimet.mpg.de)

**Abstract.** When a snowball Earth deglaciates through a very high atmospheric $CO_2$ concentration, the resulting inflow of freshwater leads to a stably stratified ocean, and the strong greenhouse conditions drive the climate into a very warm state. Here, we use a coupled atmosphere-ocean general circulation model, applying different scenarios for the evolution of atmospheric $CO_2$, to conduct the first simulation of the climate and the three-dimensional ocean circulation in the aftermath of the Marinoan
snowball Earth. The simulations show that the strong freshwater stratification breaks up on a timescale in the order of $10^3$ years, mostly independent of the applied $CO_2$ scenario. This is driven by the upwelling of salty waters in high latitudes, mainly the northern hemisphere, where a strong circumpolar current dominates the circulation. In the warmest $CO_2$ scenario, the simulated Marinoan supergreenhouse climate reaches a global mean surface temperature of about 30°C under an atmospheric $CO_2$ concentration of $15 \times 10^3$ parts per million by volume, which is a moderate temperature compared to previous estimates.
Consequently, the thermal expansion of seawater causes a sea-level rise of only 8 m, with most of it occurring during the first 3000 years. Our results imply that the surface temperatures of that time were potentially not as threatening for early metazoa as previously assumed. Furthermore, the short destratification timescale found in this study implies a very rapid accumulation of Marinoan cap dolostones, given that they were deposited in a freshwater environment.

## 1 Introduction

We apply a coupled atmosphere-ocean general circulation model (AOGCM) to study the transient period after the deglaciation of the Marinoan snowball Earth, including, for the first time, the three-dimensional ocean circulation. In contrast to the well studied snowball Earth climate and its initiation (e.g. Poulsen and Jacob, 2004; Voigt et al., 2011; Fiorella and Poulsen, 2013; Abbot et al., 2012, 2013), the processes during the supergreenhouse aftermath are much less understood. Cap dolostone formations show signs of rapid accumulation (Allen and Hoffman, 2005; Hoffman, 2011), but also hold magnetic reversals in-
dicating much longer accumulation times (Trindade et al., 2003; Font et al., 2010). At the same time, the hot climate, together with the physical and geochemical state of the ocean, could be severe for early metazoa, which possibly developed prior to the Marinoan snowball Earth (Dohrmann and Wörheide, 2017; Turner, 2021). For a better understanding of Earth's geological and biological record, an improved knowledge about the climate after the Marinoan snowball Earth is necessary.





The Marinoan snowball Earth was terminated around 635 million years ago (Ma) and had a duration of 5–15 million years
(Kendall et al., 2006; Calver et al., 2013; Prave et al., 2016). During the globally frozen state kilometer-scale continental ice
sheets and a several hundred meter, up to one kilometer, thick layer of sea ice formed (Hoffman, 2011; Abbot et al., 2013).
Beneath this global ice cover the ocean was hypersaline, geochemically evolved through ridge volcanism and well mixed
(Le Hir et al., 2008b; Gernon et al., 2016; Ashkenazy et al., 2013). The subsequent transition from the cold snowball to a warm
greenhouse climate was rapid and globally synchronous, as can be inferred from the sharp contact between the glacial deposits
of the panglacial state and the overlying carbonate formations (Kennedy, 1996; Hoffman et al., 1998; Calver et al., 2013;
Hoffman, 2011). The deglaciation causes a strong stratification in which the freshwater of the molten ice caps overlies the cold
and salty waters of the snowball Earth ocean (Shields, 2005). The stratification is then further increased by the rapid warming
of the surface layer, as a strong greenhouse climate develops. The partial pressure of $CO_2$ is expected to have reached 0.01-0.1
bar at the end of the Marinoan snowball Earth (Kasemann et al., 2005; Le Hir et al., 2008c; Abbot et al., 2012), promoting very
high temperatures in its aftermath.

In this study, we simulate the full transition from a globally frozen ocean into a supergreenhouse climate with an AOGCM
including the Marinoan topography. The focus will be on the evolution of the global parameters of climate and ocean circula-
tion. Yang et al. (2017) use a one-dimensional vertical mixing model to provide a first estimate of the destratification timescale
of a strongly stratified ocean after a snowball Earth. However, including a temporally and spatially variable surface forcing
and three-dimensional ocean dynamics can influence the outcome substantially. Furthermore, we conduct a set of sensitivity
experiments, encompassing scenarios from very fast to no atmospheric $CO_2$ removal, to acknowledge the uncertain temporal
evolution of the atmospheric $CO_2$ concentration (Le Hir et al., 2008a). Hence, we aim to give a first order estimate of possible
scenarios for the ocean circulation and the prevailing climatic conditions in the aftermath of the Marinoan snowball Earth.

In the following section we describe the model used in our study and the adaptions that were made to create the Marinoan
setup. Section 3 then gives an overview over the experimental strategy, and in Sec. 4 we describe characteristics of the Marinoan
control climate and the transition into and out of the snowball Earth. The transient response of the ocean in the aftermath of
a snowball Earth is presented in Sec. 5, and the ensuing supergreenhouse climate is described in Sec. 6. We discuss the
implications of our findings with respect to the climatic conditions and the ocean destratification process in Sec. 7, before a
conclusion is provided in Sec. 8.

## 2 Methods

### 2.1 Model

We use the icosahedral nonhydrostatic Earth system model (ICON-ESM), which couples the atmosphere general circulation
model (AGCM) ICON-A to the ocean general circulation model (OGCM) ICON-O through the YAC coupler (Hanke et al.,
2016). A specialty of the ICON model family is the unstructured grid consisting of triangles with quasi-uniform cell area. This
allows us to flexibly adapt boundary conditions as the convergence of longitudes at the poles has no impact on the grid.





Giorgetta et al. (2018) and Crueger et al. (2018) give a detailed description of the atmospheric component ICON-A, and the implementation of the nonhydrostatic dynamical core on the icosahedral grid is described in Zängl et al. (2015). The atmosphere grid used in this study has a nominal mean horizontal resolution of 315.6 km and is divided into 47 levels reaching to a height of 83 km. The vertical spacing of levels increases with height, and lower levels follow the topography (Giorgetta

et al., 2018). ICON-A has inherited its physics, that is, the parameterization of physical processes, from the well established AGCM ECHAM (Giorgetta et al., 2013) which, on its own or as part of a coupled AOGCM, has already been used in earlier studies on the snowball Earth (Marotzke and Botzet, 2007; Voigt and Marotzke, 2010; Voigt et al., 2011; Abbot et al., 2012).

The implementation of the primitive equations of the ocean on the ICON grid is described in Korn (2017). The nominal mean horizontal resolution of the ocean grid is 157.8 km with 35 vertical levels of increasing thickness with depth. The model

uses a rescaled vertical "z*" coordinate that follows the surface elevation and allows sea ice to become thicker than the first ocean layer (Campin et al., 2008). Eddy-induced transport is parameterized following Gent et al. (1995), and isopycnal mixing is described based on the formulation by Redi (1982). Both apply a constant diffusion coefficient of 1000 m$^2$ s$^{-1}$. For vertical mixing we use the approach of Gaspar et al. (1990), which relates the vertical diffusivity parameter to the turbulent kinetic energy (TKE). The resulting vertical diffusivities are inversely proportional to the Brunt-Vaïsala frequency, so that the scheme

is accounting for the inhibiting effect of stratification on vertical mixing. The parameters of this scheme are calibrated in order to achieve a good ocean circulation in a pre-industrial control simulation. Thereby, the minimum available TKE $\overline{e}_{min}$ and the calibration constant $c_k$ are increased by factors of four and five respectively, compared to the values in Gaspar et al. (1990).

The freezing point of seawater is -1.8°C in our model, and sea ice is assumed to have a constant salinity of 5 psu. Sea-ice dynamics are included in the control simulation and during the snowball initiation as well as the greenhouse climate after the

snowball Earth. However, they are turned off during the period of global glaciation, because the current implementation of sea-ice dynamics is not suited for simulating the dynamics of a globally frozen ocean with large gradients in ice thickness. Sea-ice thermodynamics follow the 0-layer formulation by Semtner (1976) which has only three prognostic variables: ice thickness, snow thickness and surface temperature. The albedo of sea ice increases nonlinearly from 0.65 at 0°C to a maximum value of 0.7 at lower temperatures. Sea ice is considered snow-covered if the water equivalent of the snow depth exceeds 0.01 m, and

the albedo of snow on sea ice increases from 0.7 at 0°C to a maximum of 0.85. The comparably high albedos of sea ice and snow follow from the formulation of sea-ice thermodynamics. They are set in a way to improve the sea ice distribution of the pre-industrial control simulation and to allow snowball initiation and melting at appropriate $CO_2$ concentrations.

The model is run with different time steps for the atmosphere, ocean and radiation components. The physical parameterizations of the atmosphere use a step size of 15 minutes, but the dynamical core performs ten substeps for every physics

calculation. The ocean component uses a time step of 60 minutes, and the coupling between atmosphere and ocean is done after every ocean time step. The radiation is calculated every 120 minutes. When the melting of the snowball Earth is initiated, the atmosphere and radiation time steps are reduced, and stratospheric damping parameters are adapted to ensure numerical stability. During this phase, the atmosphere component uses a time step of 12 minutes with 12 substeps of the dynamical core, and the radiation is calculated every 48 minutes. The ocean and coupling time steps are set to 48 minutes accordingly. Once

sea ice has retreated and the model has warmed sufficiently, time steps and damping parameters are set back to normal values.





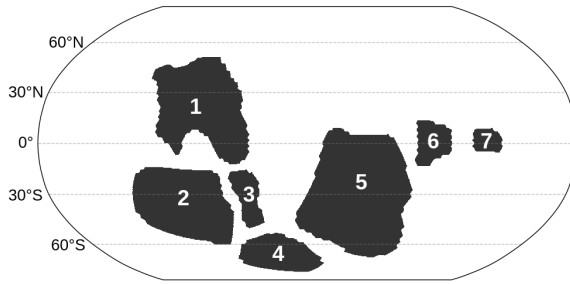

**Figure 1.** Topography of the Marinoan setup. Land areas (dark grey) have an elevation of 300 m above sea level and the ocean (white) has a depth of 3500 m. The distribution of continents follows Li et al. (2013) and Merdith et al. (2017) and is simplified in a sense that closely located cratons are summarized in larger continental areas. The continents include the following cratons, 1: Australia, India and other, 2: Congo and other, 3: Kalahari and Rio Plata, 4: West Africa and other, 5: Laurentia and other, 6: Siberia, 7: North China.

The performance of ICON-ESM in a higher resolution setup has recently been evaluated by Jungclaus et al. (in review). The lower resolution used here produces similar but generally larger biases when simulating a pre-industrial climate. Characteristics of our pre-industrial control simulation are a strong overturning circulation and warm biases in upwelling regions and the Southern Ocean.

## 2.2 Marinoan Setup

Based on the reconstructions of Li et al. (2013, Fig. 6) and Merdith et al. (2017, Fig. 11,12), we adapted the distribution of continents for the time of the Marinoan snowball Earth (Fig. 1). Continents are set to have a flat topography of 300 m above sea level. The ocean has a uniform depth of 3500 m, so that the total ocean volume is similar to the volume of the present-day oceans. This simplified topography is a first order approximation of the Marinoan conditions, but it is sufficient for the focus 100 of this paper, which is on global transitions of the climate and the large scale ocean circulation

The land surface has uniform values with no vegetation, glaciers or lakes prescribed. Soil parameters are close to the values of sandy loam, and the surface albedo of land is chosen in a way that the background albedo of the Earth's surface is close to its present-day value. River runoff is distributed over all coastal grid cells weighted by latitude, with maximum runoff at the equator.

We use a linear parameterization of ozone photochemistry (Cariolle and Teyssedre, 2007), which allows the distribution of ozone to follow the changing height of the troposphere. No aerosols are prescribed, and a constant orbit of the year 1850 C.E. is used, similar to the pre-industrial control simulation. Total solar irradiance is reduced to 95% of the present-day value to account for the weaker sun of the late Neoproterozoic (Gough, 1981). Water vapour and $CO_2$ are the only greenhouse gases, and the atmospheric $CO_2$ concentration is set to 1500 parts per million by volume (ppmv) in the control run, to create a control 110 state with a global mean temperature comparable to the pre-industrial climate.





**Table 1.** Prescribed $CO_2$ forcings for the set of simulations presented in this study. In scenario SC-CONST the $CO_2$ concentration is reduced stepwise to 18, 17, 16 and 15 $\times 10^3$ ppmv in years 4800, 6400, 7600 and 7940 to avoid model instability at too high temperatures. Similarly, there are stepwise reductions to 17 and 15 $\times 10^3$ ppmv in years 4600 and 7290 in SC-CONST-TKE.

| name | years | $CO_2$ concentration |
|------|-------|----------------------|
| CONTROL | 0–4000 | 1500 ppmv |
| INIT | 4000–4300 | 10 ppmv |
| MELT[a] | 4300–4600 | $20 \times 10^3$ ppmv |
| SC-CONST | 4600–10,500 | $20$–$15 \times 10^3$ ppmv |
| SC-SLOW | 4600–10,500 | 0.05% annual decay |
| SC-FAST | 4600–7000 | 1% annual decay |
| SC-CONST-TKE[a] | 4500–10,500 | $20$–$15 \times 10^3$ ppmv |

[a]The salinity field is simplified to a two layer case in year 4500

The Marinoan setup is initialized from a state of an earlier model version that included shelf regions and was spun-up for 3000 years starting from a homogeneous state at 5°C and a salinity of 34.3 psu. After the adaption of the ocean bottom topography, the model is run for another 1000 years until a stable climate is reached. The last 100 years of this simulation serve as the Marinoan control climate.

## 3 Experimental Design

Our experiments are a continuous simulation from a control climate into a snowball Earth and subsequently into a strong greenhouse climate. Both transitions are induced through modulating the atmospheric $CO_2$ concentration. All simulations are summarized in table 1.

Starting from the end of the control simulation, the $CO_2$ concentration is reduced to 10 ppmv in INIT to create a global
glaciation. In order to achieve a sea-ice thickness realistic for a "hard" snowball Earth, it would be necessary to integrate the model over several thousand years, and the following melting period would similarly require a very long simulation. Additionally, in our model setup no equilibrated snowball state could be achieved due to the missing implementation of geothermal heat flux. Therefore, we instead let the model create an only moderately thick layer of sea ice within 300 years, and already then initiate the deglaciation through an increased $CO_2$ concentration of $20 \times 10^3$ ppmv in MELT.

An atmospheric $CO_2$ concentration of $20 \times 10^3$ ppmv is within the $10^4$–$10^5$ ppmv range of estimates for the aftermath of the Marinoan snowball Earth (Kasemann et al., 2005; Bao et al., 2008; Abbot et al., 2012). However, it is important to stress that this value is mainly a feature of the formulation of sea-ice thermodynamics and the albedos of ice and snow in our model. The impact of these parameters on snowball Earth dynamics was extensively discussed in the literature (Lewis et al., 2006; Abbot et al., 2010; Abbot and Pierrehumbert, 2010), and a much more sophisticated sea-ice and land glacier model is needed for a
reliable estimate of the snowball Earth deglaciation threshold (Abbot et al., 2010).





Two hundred years after the onset of deglaciation, when sea-ice extent has retreated to about 35% of the ocean area, the salinity distribution is modified manually to represent a much stronger stratification: In the first 23 levels, containing the upper 1020 m of the water column, salinity is set to a uniform value of 5 psu, whereas in the 12 lower levels, or 2480 m, the concentration is set to about 46.7 psu. Here, the salinity of the freshwater layer corresponds to the fixed salinity of sea ice

used in our model, and the salinity of the brine layer is calculated so that the total amount of salt in the ocean stays the same. The distribution of temperature and all other parameters remains unchanged. This procedure avoids long integration times and potential numerical instabilities caused by large gradients in ice thickness.

Benn et al. (2015) estimate the volume of continental ice sheets to be around 170 million $km^3$ for $CO_2$ concentrations below $20 \times 10^3$ ppmv, which would translate into an oceanic freshwater layer of roughly 400 m thickness. This is 40% of the thickness

of the freshwater layer applied in our study, and the freshwater amount would be even smaller for a higher $CO_2$ deglaciation threshold (Benn et al., 2015). Adding to that the uncertain freshwater amount of roughly 500–1500 m coming from melting sea ice (Tziperman et al., 2012; Abbot et al., 2013), we argue that the freshwater layer thickness used in our study is plausible, though potentially at the lower end of possible thicknesses. As a consequence, the simulated destratification timescales could indeed be prolonged if the initial amount of freshwater was larger, and the continuous melting of land glaciers would keep

surface salinities low for a longer time. However, this is counteracted by the two-layer assumption of the salinity field, which is a highly idealized scenario and more extreme than a possible stratification after the Marinoan snowball Earth. A less extreme scenario for the vertical salinity distribution could reduce the stratification timescale again.

After the adaption of the salinity distribution, the deglaciation run MELT is continued for another 100 years before the simulation is divided into three different scenarios. These scenarios are chosen to cover a broad range of possible evolutions of

the atmospheric $CO_2$ concentration, going from no removal to an extremely rapid removal of $CO_2$. This procedure is motivated by the unknown contribution of the oceanic $CO_2$ uptake, which, in contrast to the long-term removal through continental weathering (Le Hir et al., 2008a), could cause a substantial reduction in the atmospheric $CO_2$ concentration on a timescale of several $10^3$ years, depending on the $CO_2$ saturation state of the subglacial ocean (Le Hir et al., 2008c).

In the first scenario ("SC-CONST") the atmospheric concentration of $CO_2$ is kept constant at a high value. This scenario

represents the case of maximum warming and no significant $CO_2$ uptake by either land or ocean. However, as the model is prone to numerical instabilities at too high temperatures, the $CO_2$ concentration is reduced in several steps from 20 to $15 \times 10^3$ ppmv. The scenario "SC-SLOW" deploys an exponential decay of 0.05 % per year and thereby emulates a modest removal of $CO_2$ from the atmosphere. This exponential decay is increased to 1 % per year in the scenario "SC-FAST" which serves as an extreme case of quick $CO_2$ removal. In SC-SLOW and SC-FAST the exponential decay of the atmospheric $CO_2$ concentration

stops when the value used in CONTROL is reached, which happens in years 9780 and 4858, respectively.

In addition to the three scenarios with distinctive $CO_2$ concentration pathways, we conduct the experiment "SC-CONST-TKE", which follows a similar $CO_2$ pathway as SC-CONST, but uses different settings in the TKE scheme. Hence, this simulation serves us to test the robustness of our results with respect to vertical mixing. In this experiment the parameters of the vertical mixing scheme are set to the standard values suggested by Gaspar et al. (1990). The minimum available TKE is

$\overline{e}_{min} = 10^{-6}$ $m^{-2}$ $s^{-2}$, and the calibration constant is $c_k = 0.1$, implying much weaker vertical mixing. We note that when we





**Table 2.** Comparison of the Marinoan control climate (CONTROL, TSI = 95 %, $CO_2$ = 1500 ppmv, $CH_4$ = 0 ppmv, $N_2O$ = 0 ppmv) and the pre-industrial control simulation with present-day continents (PI, TSI = 100 %, $CO_2$ = 284 ppmv, $CH_4$ = 0.808 ppmv, $N_2O$ = 0.273 ppmv). Both columns represent data averaged over 100 years.

|  | CONTROL | PI |
|---|---|---|
| GSAT (°C) | 13.4 | 13.9 |
| mean sea surface temperature (°C) | 15.3 | 19.1 |
| mean sea surface salinity (psu) | 34.2 | 34.6 |
| mean ocean potential temperature (°C) | 1.9 | 3.8 |
| sea-ice extent ($10^6$ km$^2$) | 23.0 | 12.5 |
| sea-ice volume (km$^3$) | 19,519 | 32,050 |

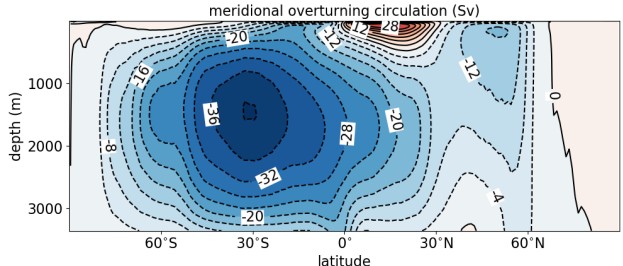

**Figure 2.** Meridional overturning circulation of the Marinoan control climate. Negative values and blue colors refer to an anti-clockwise circulation.

use the smaller values of Gaspar et al. (1990) in a simulation with present-day continents, it leads to a complete breakdown of the Atlantic Meridional Overturning Circulation, and those parameters were specifically increased to reduce the bias between observations and the model simulation. SC-CONST-TKE starts at the time where the salinity distribution is adapted in MELT and is, similar to the other scenarios, stopped when the climate approaches an equilibrium.

## 4   Control Climate and Snowball Period

Since the Neoproterozoic atmospheric $CO_2$ concentration is only loosely constrained (Hoffman et al., 1998), we here choose 1500 ppmv for the Marinoan control climate, because this leads to a global mean 2 m air temperature (from here on referred to as GSAT) comparable to that of the pre-industrial control simulation, despite the weaker solar forcing. While the GSAT of the Marinoan control climate is similar to the pre-industrial one, sea surface temperatures are substantially lower (Tab. 2). This is a consequence of the larger continental area in low latitudes. Furthermore, deep ocean temperatures are lower in the Marinoan control climate than in the pre-industrial simulation, causing a weaker resistance towards global glaciation due to the smaller thermal inertia. The deep waters of the ocean are formed at the south pole, where the waters sink to the ocean bottom during





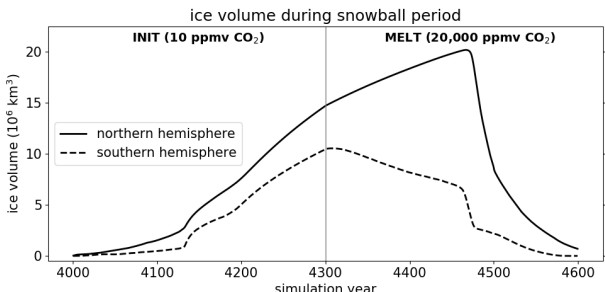

**Figure 3.** Evolution of the sea-ice volume during the snowball period, divided into the northern (solid) and the southern (dashed) hemisphere. The vertical line denotes the transition from a low $CO_2$ concentration in INIT to a high $CO_2$ concentrations in MELT.

the aureal winter. Additionally, convection occurs in both hemispheres along the sea-ice boundary. In the northern hemisphere there are no continents north of 51°N, leading to an extended sea-ice area with large seasonal fluctuations. The meridional

overturning circulation (MOC, Fig. 2) is characterized by a large cell with deep waters flowing northward along the ocean bottom from the deep water formation regions close to the south pole to a circumpolar current at 60°N. The single-celled shape of the Marinoan MOC is different from that of the present-day MOC, which consists of two counter-rotating cells. This is because in the present-day oceans the deep waters are formed in both hemispheres and with different density properties. In contrast to that, the location of continents in the Marinoan setup seems to favor deep water formation only at the south pole,

and the flat ocean bottom allows those waters to penetrate far into the northern hemisphere, hindering the formation of deep waters there.

     The northern circumpolar current of the Marinoan control climate has a transport of around 960 Sv and is therefore an important component of the ocean circulation. The relative strength of this flow, compared to the present-day Antarctic Circumpolar Current (ACC), is a consequence of the missing continental barriers north of 51°N and the weak bottom drag due

to the flat ocean bathymetry. In a more realistic setup, including bathymetric features and a higher horizontal resolution, the circumpolar current would be slowed down substantially by topographic form stress and the downward momentum transport through eddies (Wolff et al., 1991). However, even though the circumpolar current is likely too strong in our simulation, the continental reconstructions of Li et al. (2013) and Merdith et al. (2017) are clear in that probably no continents were located north of 60°N. The existence of a circumpolar current, potentially stronger than the present-day ACC, is therefore conceivable.

We will return to the influence of the circumpolar current on our model results during the discussion in Sec. 7.3.

     After the initial reduction of the atmospheric $CO_2$ concentration to 10 ppmv in run INIT, it takes 134 years until the sea ice covers more than 95% of the ocean surface area. The sea ice then quickly grows in vertical direction (Fig. 3), and at the end of the 300-year long freezing period it reaches a global mean ice thickness of 65.6 m. The thickest sea ice can be found at the

poles with 84 m in the southern hemisphere and up to 102 m in the northern hemisphere, while the thinnest ice is located in the subtropics and tropics with thicknesses between 50 m and 65 m in most parts.





When the atmospheric $CO_2$ concentration is increased in MELT, the global ice volume initially continues to grow in the northern hemisphere. A strong asymmetry between the hemispheres develops, with decreasing ice thickness in the subtropics of the southern hemisphere, mainly next to the western coast of the continents, and increasing or constant ice thickness in other regions. In year 4463, 163 years after the start of MELT, sea-ice cover falls below 95% of the ocean area and a rapid deglaciation sets in.

In year 4500 of MELT, when the salinity distribution of the water column is adapted to the two-layer case described in Sec. 3, sea-ice cover has retreated to 34% of the ocean area and GSAT has reached 2.5°C. The adaption of the salinity field slows down the warming and the ice retreat in the first years, but then leads to a faster warming of the surface layer, as less cold waters reach the surface from the deep. One hundred years later, at the end of MELT, GSAT has already reached 22.6°C, and sea-ice cover has decreased to less than 5%.

## 5  Temporal Evolution of Climate and Ocean Circulation in the Snowball Earth Aftermath

We now provide a description of the transient response of the climate and the ocean circulation to the different scenarios. Naturally, a $CO_2$ pathway that keeps the greenhouse conditions intact for a longer time will cause more warming than a fast $CO_2$ decrease, which is apparent in Fig. 4 (a). As the climate is already very warm at the end of MELT in year 4600, the start of the 1% annual decay scenario SC-FAST causes the surface to start to cool within a few decades. From there on, a rapid cooling takes place and the surface temperature reaches the value of the control climate within 400 years from the start of SC-FAST, approximately 140 years after the $CO_2$ concentration reached the value of the control simulation. The surface then cools further, because the continuously low surface salinity promotes the formation of sea ice and the deep ocean is still cold. By contrast, the annual $CO_2$ reduction in SC-SLOW is much smaller. Here, the surface climate continues to warm for 160 years before the $CO_2$-induced cooling becomes visible. The warming of the global ocean persists for almost 2000 years. In SC-CONST the four individual $CO_2$ reductions, necessary to keep the model numerically stable, are visible as sudden drops in the surface temperature evolution. The first reduction in year 4800 stops the initial strong warming trend and a slow surface cooling sets in. However, the deep ocean keeps heating up, which causes a second phase of surface warming between years 5600 and 7600, where at the end the $CO_2$ concentration had to be reduced again. Afterwards, the warming of the surface, as well as the deep ocean, continues at a slower, decelerating pace.

The sea-level change due to the thermal expansion of the warming ocean largely follows the evolution of the mean ocean temperature (Fig. 4 (b)). As our model does not include ice sheets on land, we cannot give an estimate about the much larger sea-level change from melting land glaciers. When in the following the terms sea-level rise or sea-level change are used, we therefore only refer to the change resulting from the thermal expansion of seawater. We derive the global mean sea-level rise on annual mean data by calculating the thermal expansion of each grid box from one year to the next using the formula from McDougall (1987). The individual values are then stacked up vertically and averaged horizontally to arrive at the global mean sea-level rise. During the 300 years of deglaciation in MELT, the thermal expansion of seawater already leads to a sea-level change of 1.2 m. In the fast $CO_2$ reduction pathway SC-FAST, however, the further sea-level change is small, as the deep ocean

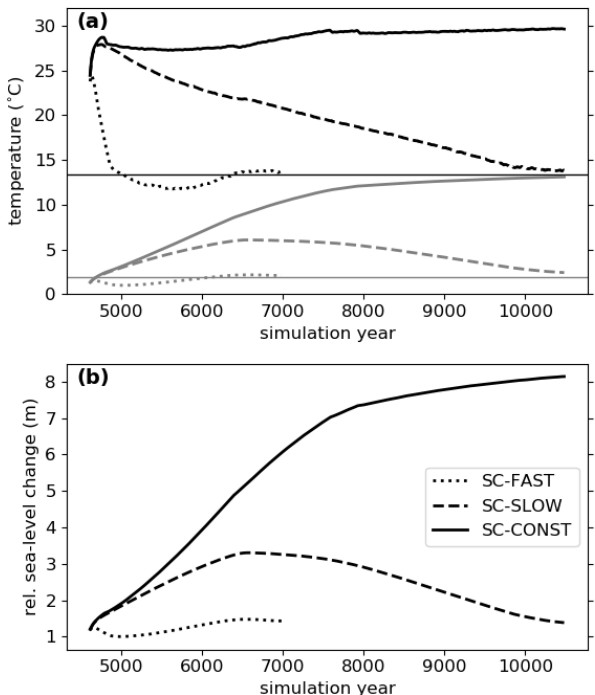

**Figure 4.** Temporal evolution of **(a)** global mean 2 m air temperature (black) and ocean mean potential temperature (grey) and **(b)** relative sea-level change due to thermal expansion with respect to the starting point of each scenario. All plots show 30 year running means. The horizontal lines depict the value of the Marinoan control climate.

only warms slightly and the surface cools again. When applying the intermediate scenario SC-SLOW, the total sea-level rise since the deglaciation reaches a maximum of 3.3 m around year 6600, before the sea level drops again. Lastly, in SC-CONST the sea level continues to rise over the whole simulation, but the speed of the increase slows down, influenced also by the stepwise $CO_2$ reductions. At the end of the simulation, this increase has added up to 8 m, and a small trend of +0.2 mm per year is still visible.

To characterize the ocean circulation and the stratification, Fig. 5 shows zonally averaged sections of the oceanic meridional overturning circulation (MOC) and the salinity field for the control climate and in different years of SC-CONST. The MOC of the control climate is dominated by a large anti-clockwise cell between 80°S and 30°N. Directly after the melting of the snowball Earth, and after the adaption of the salinity field, this dominant cell is gone in all scenarios. Instead, there are distinct cells in the freshwater and the brine layer. The stratification starts to break up in the high latitudes, driven by local overturning

cells at the poles and the continuously strong circumpolar current in the northern hemisphere. All scenarios show a similar initial evolution of the MOC in the first 1000 years after the deglaciation and ultimately a recovery of the dominant anti-clockwise cell (Fig. 6(a)). This recovery is rapid and occurs over a few hundred years in SC-FAST; it is slower in SC-SLOW,





**Figure 5.** Left: Zonally averaged streamfunction of the Marinoan control climate and at five representative times during the strong warming scenario SC-CONST. Positive values depict a clockwise circulation. Right: Zonally averaged salinity field for the same years. The data of both columns represents 30 year averages around the given timestamp.

where there is an overshoot in the overturning strength before the circulation approaches the control state. In SC-CONST the MOC recovers gradually over a timescale of more than 2000 years and never reaches the strength of the control climate.

Circumpolar currents are another important feature when considering ocean destratification, as they are associated with exchange between surface waters and the deep ocean through deep Ekman cells. Figure 5 shows that the largest initial deviation from the horizontally layered salinity field is through upwelling of salty waters northwards of 60°N, which occurs in all scenarios. To quantify the strength of the circumpolar current located there, we calculate the flux through a meridional section





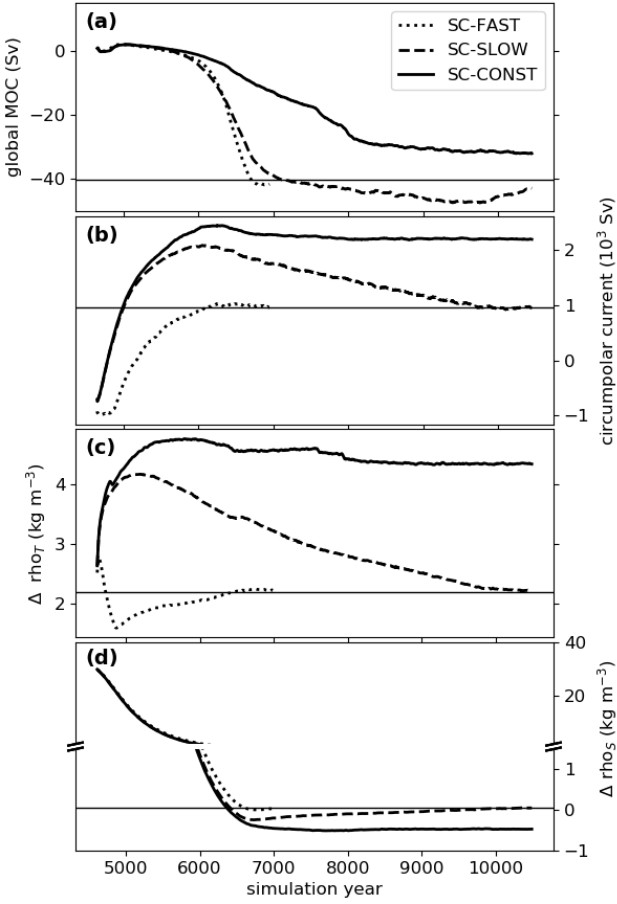

**Figure 6.** Temporal evolution of the large scale ocean circulation. **(a)** Strength of the zonally averaged MOC at 30°S and 1365 m depth. **(b)** Mass flux of the depth-integrated circumpolar current with positive values referring to an eastward flow. **(c)** Temperature contribution to the density difference from the ocean bottom to the surface. Higher values correspond to a stronger stratification. **(d)** Same as in (c), but for salinity; note the different scaling above and below the break in the y-axis. The plots show 100 year running means. The black horizontal lines depict the mean value of the Marinoan control climate.

between the north pole and the northern tip of the conglomerate combining the Australian, Indian and other cratons (Fig. 6
(b)). After an initial phase of about 500 years, where the model adapts to the idealized two-layer salinity field, the circumpolar current in SC-FAST approaches the strength it has in the control climate. It accelerates much quicker in SC-SLOW and SC-CONST, where the higher surface temperatures cause a strengthening and a shift of the westerly winds to higher latitudes, intensifying the surface drag (described in Sec. 6). We elaborate on the implications that an overestimation of the current strength has on the inferred destratification timescales in Sec. 7.3.



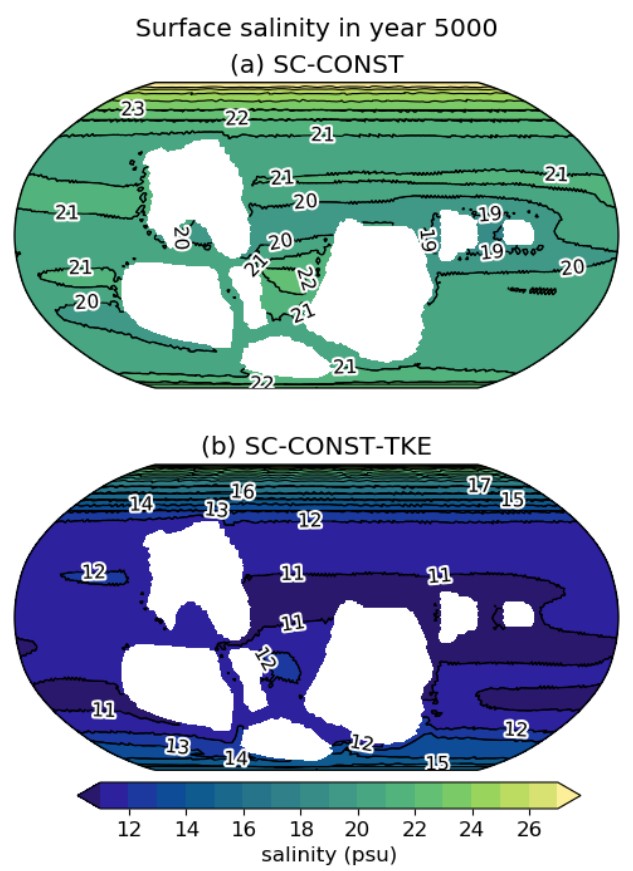

**Figure 7.** Surface salinity in year 5000 of (a) SC-CONST and (b) SC-CONST-TKE. The latter uses the parameters of the TKE vertical mixing scheme as they were suggested in Gaspar et al. (1990). The data of both plots represent a 30 year mean around the specified year.

Next, we investigate the impact of the $CO_2$ pathway on the ocean stratification. As a first order approach, the strength of the ocean stratification can be determined from the density difference between the surface and the ocean bottom. Therefore, we calculate the horizontal means of salinity and temperature in the surface and bottom layer of our model. These values can then be used to divide the stratification into a thermal and a haline component, which are shown in Fig. 6 (c) and (d). The thermal component should follow the $CO_2$ pathway more directly, while the haline component could give insight into whether certain

$CO_2$ pathways accelerate or decelerate the break-up of the imposed freshwater stratification. At the start of the scenarios in year 4600, the haline stratification is more than ten times stronger than the thermal component, with values of 30 and 2.3 kg m$^{-3}$, respectively. The thermal component of the stratification then closely follows the evolution of the surface temperature (see Fig. 4 (a)), with large differences between the scenarios and the strongest stratification in SC-CONST. In contrast to that, the haline stratification drops from 30 to 0.8 kg m$^{-3}$ within 1500 years from the start of the scenarios and is mostly independent of the

applied $CO_2$ pathway.





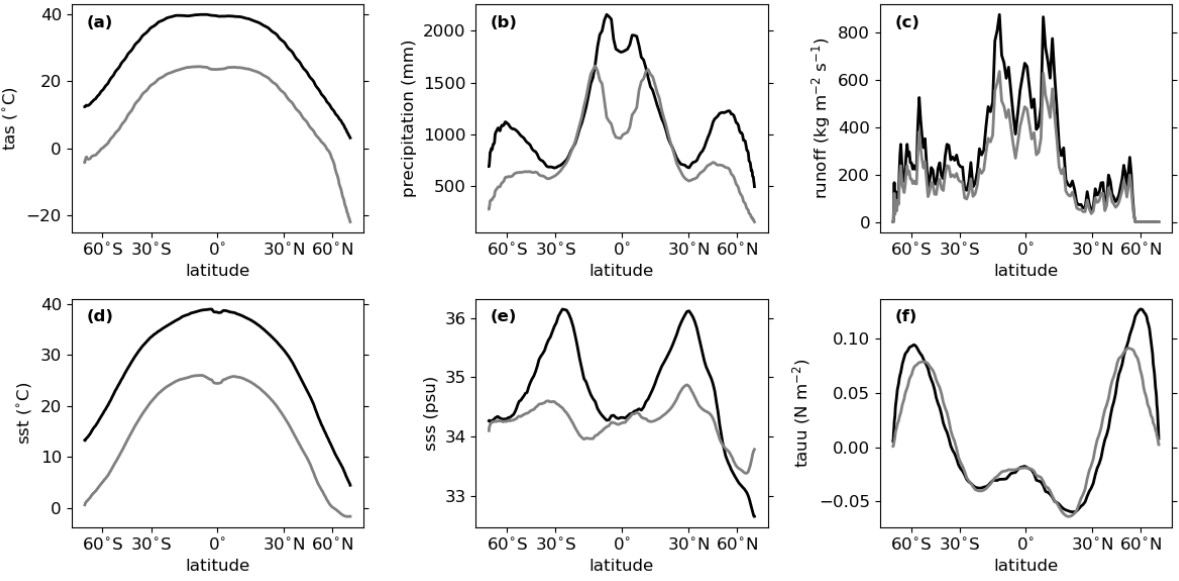

**Figure 8.** Zonal averages of the supergreenhouse climate in SC-CONST (black) and the Marinoan control climate (grey) for **(a)** 2 m air temperature, **(b)** precipitation, **(c)** hydrological discharge from land grid cells, **(d)** sea surface temperature, **(e)** sea surface salinity and **(f)** zonal windstress.

Lastly, we quantify the impact of the parameterized vertical mixing on the destratification timescale through the simulation SC-CONST-TKE. This simulation follows a similar $CO_2$ pathway as SC-CONST, but applies much weaker vertical mixing, as described in Sec. 3. Figure 7 shows the surface salinity 500 years after the insertion of the idealized salinity field for the simulations SC-CONST in (a) SC-CONST-TKE in (b). Indeed, surface salinities in that year are approximately 10 psu smaller in SC-CONST-TKE than in SC-CONST, but also here the stratification break-up is driven from the high latitudes, and SC-CONST-TKE reaches surfaces salinities of 20 psu only about 500 years later than SC-CONST. Hence, a weaker parameterized vertical mixing can partly delay the stratification break-up, but it does not lead to a qualitatively different behavior, in which the stratification would persist for $10^4$ years or more.

## 6   Supergreenhouse Climate

The deglaciation of a snowball Earth applies a strong evolutional pressure because it encompasses a very rapid transition between two climate states, but also because the ensuing warm climate could be threatening for some early metazoa. A scenario with long-lived extreme greenhouse conditions is therefore of particular interest, to understand how severe the climate could have possibly been in the aftermath of the Marinoan snowball Earth. In this section, we describe the supergreenhouse climate of the $CO_2$ scenario that gives the strongest warming in our simulations. For that we evaluate the mean climate of the last 100 years of SC-CONST, which represents a quasi-stable state under a $CO_2$ concentration of $15 \times 10^3$ ppmv.





The annual global mean 2 m air temperature of the simulated supergreenhouse climate is 29.7°C, but there exist large regional and temporal variations. While the mean temperature reaches 50°C over some continental areas in the subtropics, mean sea surface temperatures are only 3°C at the north pole. Daily temperature averages can be as high as 68°C, while in other areas temperatures below -5°C are possible. In general, the meridional temperature distribution shows the same pattern as in the control climate, shifted towards higher values (Fig. 8 (a), (d)).

Although GSAT is more than 16°C higher than in the control climate, precipitation and continental runoff are only increased by 37% and 38%, respectively, which is a slightly larger increase than found by Le Hir et al. (2008a). The main precipitation increase compared to the control climate occurs in the tropics and the mid to high latitudes, so that the general pattern of high precipitation in these regions and low precipitation in the subtropics is strengthened.

One of the reasons for the existence of still comparably cold regions in the Marinoan supergreenhouse climate is the circumpolar current in the northern hemisphere. Through the poleward shift of strengthened westerly winds (Fig. 8 (f)), the circumpolar current has a water transport of 2190 Sv in the supergreenhouse climate, which is more than two times stronger than in the control climate. As discussed in Sec. 4, the remarkable strength of the circumpolar current is mainly a consequence of the simplified ocean bathymetry in our setup, but the general existence of a circumpolar current is likely. Furthermore, that a circumpolar current strengthens in scenarios of increased greenhouse warming is known also from the warming of the present-day climate (Thompson and Solomon, 2002; Russell et al., 2006). The strong circumpolar current prevents warm water masses from penetrating into higher latitudes. As a consequence, sea surface temperature and salinity are both substantially lower in the high latitudes of the northern hemisphere than in those of the southern hemisphere (Fig. 8 (d), (e)).

# 7 Discussion

## 7.1 Impact of CO$_2$-Scenarios

We included three different pathways for the atmospheric CO$_2$ concentration after the deglaciation, ranging from a very fast decline to no decline of the strong greenhouse conditions. The most straightforward difference between the scenarios is in the evolution of the global mean 2 m air temperature and consequently the mean ocean temperature and the associated sea-level rise. The scenario SC-CONST shows a permanent warming and a sea-level rise of up to 8 m, in SC-SLOW the surface warms for only a few hundred years, but the ocean temperature continues to increase for around 2000 years, and in SC-FAST the climate quickly approaches the control climate without any major sea-level rise (see Fig. 4).

While the direct response of the surface temperature and the delayed response of the sea-level rise are readily derived differences coming from the implied different greenhouse forcing in the scenarios, the connection of the ocean circulation to the CO$_2$ pathways is less clear. The strength of the circumpolar current is shown to depend on the scenario (Fig. 6 (b)), with larger transports in warmer climates. This follows from the poleward shift of westerly winds under global warming, as described in Sec. 6. Furthermore, the meridional overturning circulation (MOC) recovers on a similar timescale in SC-FAST and SC-SLOW, but much slower in SC-CONST (Fig. 6 (a)). In the first thousand years after the deglaciation, the MOC is mostly gone in all scenarios, because the surface freshwater layer inhibits deep water formation. This haline stratification breaks up on





the same timescale in all scenarios, but in SC-CONST a continuously strong thermal stratification limits the full recovery of the

MOC (Fig. 6 (c), (d)). The independence of the haline destratification timescale on the $CO_2$ pathway leads to the conclusion

that it is not the MOC that is driving the break up, as the slower MOC recovery in SC-CONST should otherwise also delay the

destratification in the salinity field. The strength of the MOC is rather a consequence of the stratification strength. We return to

the question of what is driving the destratification in Sec. 7.3.

The procedure of applying different $CO_2$ pathways was motivated by the unknown carbon content of the ocean at the end

of the deglaciation and the resultant possibility to either take up atmospheric $CO_2$ very quickly or not at all. It helps to test the

robustness of the simulation results with respect to the greenhouse forcing and to give a range of possible climatic evolutions.

However, prescribing the atmospheric $CO_2$ concentration neglects the interplay between the oceanic and atmospheric carbon

reservoirs, and for future studies a model including an interactive carbon cycle component could help to better understand the

climate after a snowball Earth.

In the following we will only discuss the warmest scenario SC-CONST, as we expect this to be the most likely case. This

assumption is based on the fact that even small areas of open ocean are sufficient for an effective dissolution of $CO_2$ in the

snowball ocean (Le Hir et al., 2008c), which means that afterwards the oceanic capacity to take up carbon is reduced strongly.

Scenarios with a smaller or faster declining $CO_2$ concentration are also less extreme and therefore less consequential for the

discussed properties. The possibility of even higher atmospheric $CO_2$ concentrations is included in the following discussion.

**7.2 Conditions After the Marinoan Snowball Earth**

With the atmospheric $CO_2$ concentration at $15 \times 10^3$ ppmv, as chosen in our scenario SC-CONST, the global mean 2 m air

temperature is simulated to be around 30°C, and large temperate areas exist in the higher latitudes. The supergreenhouse

climate simulated in this study therefore probably does not represent a major restriction for early eukaryotic life (Rothschild

and Mancinelli, 2001). With tropical sea surface temperatures of only up to 40°C, compared to previous estimates of 50–60°C

(Yang et al., 2017), it is less clear whether cyanobacteria would indeed outcompete algea in the tropics and cause the observed

"algal gap", as proposed by Brocks et al. (2017). However, temperatures could be considerably higher when pCO$_2$ was at the

upper end of the range of $10^4$–$10^5$ ppmv, which was estimated for the aftermath of the Marinoan snowball Earth (Kasemann

et al., 2005; Bao et al., 2008; Abbot et al., 2012). To determine the possible GSAT at a $CO_2$ concentration of $10^5$ ppmv, we

calculate the Marinoan equilibrium climate sensitivity from the temperature difference between the supergreenhouse climate

of SC-CONST and the Marinoan control simulation. Thereby, we arrive at a temperature increase of 4.9 K per doubling of

the $CO_2$ concentration. Using this value, we can estimate that the GSAT at $10^5$ ppmv of $CO_2$ would be around 43°C. In such

a climate, the tropical ocean would likely be dominated by cyanobacteria, but it is conceivable that polar regions still exhibit

favorable conditions for early metazoa.

The sea-level rise due to thermal expansion is strongly related to the surface warming. It contributes to the overall sea-level

change and can therefore influence the deposition of Marinoan cap dolostones. In our simulation SC-CONST, the thermal

expansion of seawater accumulates to a sea-level rise of 8 m from the start of the deglaciation, and around 90% of it occurs

within the first 3000 years. This means it mostly occurs on the same timescale as the much stronger sea-level rise due to the





melting of continental ice sheets (Hyde et al., 2000), which is 1–2 orders of magnitude larger (Benn et al., 2015). Furthermore, a sea-level rise of 8 m is considerably smaller than the 40–50 m derived in Yang et al. (2017). This difference can be attributed

to the much higher surface temperatures in Yang et al. (2017), causing the deep ocean to arrive at a potential temperature of 42°C, compared to 13°C in our setup. It can be argued that a deep ocean temperature of 42°C is too high, as it would probably require mean surface temperatures which are too high for the estimated possible range of $CO_2$ concentrations (discussed above). Our results therefore indicate that the thermal expansion of seawater is less important than it was previously assumed for the sea-level rise in the aftermath of a Marinoan snowball Earth.

Liu et al. (2014) show that isotope compositions of carbonate formations in South Australia and Mongolia indicate a deposition in two chemically distinct fluids: a plume of freshwater overlying the salty waters of the snowball ocean. The simulations presented in this study, however, show that a surface layer of low salinity is rapidly removed by the ocean circulation. The short destratification time could indeed be prolonged by the inflow of freshwater from the melting continental ice-sheets, which would keep surface salinites low, especially near coastal areas, where cap dolostones were formed (Allen and Hoff-

man, 2005; Myrow et al., 2018). Still, the rapid removal of the freshwater stratification by the ocean circulation reduces the potential timescale for cap dolostone deposition to the duration of the deglaciation of continental ice sheets, a few kiloyears (Hyde et al., 2000), given that they were deposited predominantly in a freshwater environment. While this is consistent with sedimentological features indicating rapid accumulation (Allen and Hoffman, 2005; Hoffman, 2011), paleomagnetic reversals found in the carbonates hint to much longer accumulation times of several $10^4$ to $10^5$ years (Trindade et al., 2003; Font et al.,

2010). A possible explanation for this inconsistency is the hypothesis that the Earth's inner core only started to solidify less than 600 Ma (Davies, 2015). This could lead to paleomagnetic reversals on much shorter timescales than today and explain those reversals found in Marinoan cap dolostones (Hoffman et al., 2017). The simulations presented in this study support the concept of very rapid accumulation of cap dolostones, based on the finding that the ocean circulation can break up most of the haline stratification within a geologically very short time. This finding is discussed in the next section in more detail.

## 7.3   Ocean Destratification and Vertical Mixing

Yang et al. (2017) use a one-dimensional vertical mixing model to estimate the destratification time of an idealized two-layer salinity field, similar to the one used in this study. They find timescales between $10^4$ and $10^5$ years, mostly depending on the vertical diffusivity of the ocean and the initial amount of freshwater. In contrast to that, our simulations indicate mixing times of just a few $10^3$ years. In this section, we discuss why our destratification times are so much shorter and which timescales

could be realistic.

The one-dimensional vertical mixing model used in Yang et al. (2017) and the AOGCM used in this study implement conceptually different mechanisms by which a vertical stratification can be removed. While the 1d-model describes all vertical motion through a vertical diffusion equation, the AOGCM attains the horizontal and vertical ocean circulation by solving the hydrostatic primitive equations on a three-dimensional grid (Korn, 2017), but it also adds vertical motion through a parame-

terization of small scale turbulent (vertical) mixing. Yang et al. (2017) argue that this mixing could be smaller in the warm snowball Earth aftermath than it is today, because of less energy input through weaker lunar tides and a weaker energy input





from wind. Even though the second point is questionable in the light of increasing surface winds in our simulations of the supergreenhouse climate (see Fig. 8 (f)), it could still be speculated that the short mixing times found with our model are a consequence of too much parameterized vertical mixing, because the mixing scheme was developed and adapted for a present-day
climate. However, our simulation SC-CONST-TKE shows that also substantially weaker parameterized vertical mixing only delays the stratification break-up by some 500 years. Therefore, the parameterization of vertical mixing is unlikely to be the source for the order-of-magnitude difference in the destratification timescale.

Is it then the three-dimensional nature of the ocean circulation that is the main driver of the fast stratification break-up? The plots in Figs. 5 and 7 show that waters of higher salinity reach the surface through the high latitudes, especially in the
northern hemisphere. Here, the strong westerly winds induce a massive circumpolar current that drives upwelling through surface divergence induced by Ekman transport. The clear correlation between the existence of a circumpolar current and the much stronger steepening of isohalines in the northern hemisphere is a convincing indicator that indeed the three-dimensional circulation, and not the small-scale vertical mixing, cause the fast break-up of the stratification. It is this crucial part of the ocean circulation that is missing in the vertical mixing model of Yang et al. (2017) and that explains a large part of the timescale
differences.

Nevertheless, the question remains, how much of the timescale difference is due to an overestimation of the circumpolar current strength in our setup. As discussed in Sec. 4, the simplified topography induces an unrealistically strong circumpolar current with a transport of 960 Sv during the control climate, which reaches up to 2190 Sv in the supergreenhouse climate after the snowball Earth. This is an order of magnitude stronger than the present-day ACC. Part of this difference can be
attributed to the location of continents during the Marinoan, which indeed favors a circumpolar current stronger than the ACC. A larger part of the difference, however, comes from the missing form drag at the flat ocean bottom, which is a typical issue in models with simplified topography (Bryan and Cox, 1972; Wolff et al., 1991). In a realistic circumpolar current, the eastward momentum imparted by the surface wind stress would be transferred downward through transient or standing eddies (Wolff et al., 1991). This slows down the circumpolar current substantially and leads to much less vertical transport in the deep Ekman
cell, hindering the break-up of the stratification. However, the eddy activity itself is associated with turbulent vertical mixing. Therefore, even though the strength of the circumpolar current could be greatly reduced when using an eddy-resolving model with appropriate bottom topography, this is not necessarily the case for the vertical mixing linked to the current. So, while our model probably overestimates the vertical mixing induced by the Ekman cell, it possibly underestimates the vertical mixing through eddy activity, especially in the simulation SC-CONST-TKE.

In summary, the discrepancy in destratification timescales between the study of Yang et al. (2017) and our results can largely be explained by the addition of ocean dynamics in our study. Although our simplified model topography favors a quick break-up of the stratification, it is still plausible that the ocean mixing timescale is much less than the $5 \times 10^4$ years that were suggested by Yang et al. (2017). The main reason for this is the following: The location of continents at the time of the Marinoan glaciation favors a strong circumpolar current, which accelerates the break-up of the stratification through a deep Ekman cell and strong
eddy activity (It should be noted that the circumpolar current is not the only region where the three-dimensional circulation will lead to vertical transports, but it is presumably the strongest contributor to the destratification). These effects are not accounted



for by the model of Yang et al. (2017), explaining their much longer mixing timescale. We propose that even when including the assumptions of Yang et al. (2017), a larger freshwater amount inserted over a few $10^3$ years and a weaker circumpolar current the three-dimensional ocean circulation can break up the freshwater stratification after a snowball Earth in less than

$10^4$ years.

## 8   Conclusions

In this study, we simulate the transient period from the deglaciation of a Marinoan snowball Earth into a warm supergreenhouse climate using a coupled AOGCM. The main findings and implications of our simulation results are summarized by the following points:

1. By including the three-dimensional ocean circulation in a study of the snowball Earth aftermath, we show that the strong haline stratification can break up within just a few thousand years, mostly independent of the path the atmospheric $CO_2$ concentration takes. This finding is robust, even with significantly reduced vertical mixing, and shows the important contribution of the dynamical nature of the ocean circulation. Our results therefore indicate that cap dolostones where deposited very rapidly during the period of deglaciation or shortly after that, given that they accumulated predominantly

in a freshwater environment.

2. In a Marinoan supergreenhouse climate with an atmospheric $CO_2$ concentration of $15 \times 10^3$ ppmv, the Earth exhibits a global mean temperature of only around 30°C, and large temperate areas exist in the high latitudes. The relative modest warming in our simulations, compared to previous estimates of GSAT of about 50°C, means that the impeding effect of temperature on eukaryotic life was only potentially severe with a $CO_2$ concentration at the upper end of the estimates

and not with the concentrations applied in this study.

3. Our simulations do not show a significant long-term sea-level rise due to thermal expansion, which is a consequence of the relatively lower surface and deep ocean temperatures, compared to previous studies (Yang et al., 2017). We therefore conclude that it was mainly the sea-level change through glacio-isostatic adjustment affecting the deposition of Marinoan cap carbonates and less the thermal expansion of seawater.

*Code availability.*   The model code of the specific setup used in this study, as well as model input, post-processed data and scripts used for the analysis and producing the figures, can be obtained from the Climate and Environmental Retrieval and Archive (CERA) of the World Data Center for Climate (WDCC) (Ramme, 2021).

## Appendix A:   Impact of Model Errors on the Simulations

The here used Earth system model ICON-ESM is a recently developed model (Jungclaus et al., in review). Even though part

of ICON's parameterizations are based on its predecessor MPI-ESM (Mauritsen et al., 2019), large parts of the code were built





from scratch. Therefore, the frequency with which coding errors are detected is somewhat higher than in more established models. After the simulations presented in this study were finished, a set of bugs was detected, which are related to energy fluxes at the surface. On the one hand, both moisture and the dry static energy of moist air were diffused by the surface fluxes, effectively counting the moisture flux twice. On the other hand, the latent and sensible heat fluxes where accidentally set to
zero when calculating the surface temperature of sea ice in the 0-layer Semtner model. In the following, we explain why these errors do not influence the outcome of our work.

    All conclusions of this paper are derived from first order components of the climate, like the broad-scale ocean circulation or the evolution of the global temperature distribution after the snowball Earth. Fixing the model errors does not change the qualitative nature of the simulation outcome. The first bug, causing the double counting of the moisture flux, potentially
alters the conditions of the supergreenhouse climate, but a 100 year long simulation, including all bugfixes and a retuning of model parameters for a good pre-industrial simulation, produces a very similar supergreenhouse state, with an almost identical meridional temperature distribution. The second model error, found in the calculation of the surface temperature of sea ice, effectively warms the ice surface and therefore leads to generally thinner sea ice. This is balanced by the high ice and snow albedos of the sea-ice scheme used in our model. Lower sea-ice albedos would be used in a model version where this bug
is fixed. Test runs show that fixing this bug and lowering the albedo leads to a similar behavior in a pre-industrial control simulation, but the deglaciation of a snowball Earth is much harder to achieve. However, shortly after the deglaciation, sea ice is gone completely, and it only recovers in SC-FAST and later in SC-SLOW, while it remains gone in SC-CONST and SC-CONST-TKE. Hence, large parts of our simulations are not affected by this second model error. It can thus confidently be said that the results and the conclusions of this work are robust with respect to both detected model errors.

*Author contributions.* LR and JM designed the study. LR conducted the model simulations, performed the analysis and prepared the first draft of the manuscript. JM supervised the study. Both authors contributed to the scientific discussion and the writing of the manuscript.

*Competing interests.* The authors declare that they have no conflict of interest.

*Acknowledgements.* We thank Chao Li for the internal review of the manuscript and Stephan Lorenz and Helmuth Haak for valuable technical support. This work was supported by the Max Planck Society for the Advancement of Science and the International Max Planck Research
School on Earth System Modelling. All model simulations and analyses were performed using resources of the German Climate Computing Center (DKRZ).



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
