# Peer review of "Climate and Ocean Circulation in the Aftermath of a Marinoan Snowball Earth"

_Climate of the Past, 2021_

## Author Comment (AC1)

Response to Anonymous Referee #1

- Referees comment
- Authors response

The motivation in terms of cap carbonate formation, effects on ocean life, comparison with previous estimates of the de-stratification timescale of the fresh water layer is very helpful and well written.

The model description is especially helpful. The authors seem to have identified all weaknesses in their experiment design, anticipated all possible caveats/criticisms and addressed them very well. As part of this they discuss the coarse atmospheric and oceanic resolution, flat ocean topography, the unavoidable arbitrariness of the vertical mixing scheme that is addressed by sensitivity experiments, inability of the model to simulate full-thickness ice layer that is addressed via a readjustment of the salinity stratification upon melting of sea-ice to 35% area extent, the inability of the model to represent sea ice/glacier dynamics for a thick ice cover (sea ice dynamics is appropriately turned off then, leaving only the thermodynamics active), our incomplete knowledge of $CO_2$ after snowball events that is addressed via 3 different sensitivity scenarios, and more.

Figure 2: perhaps show also the overturning for the fully glaciated state (and the temperature and salinity for both states), so that it can be compared with the following simulated times shown later.

We added the requested panels to figure 2 and modified it to have the same style as figure 5 for an easy comparison. We also removed the uppermost row in figure 5, which showed the state of the control climate again, as this information is now fully included in figure 2. We adapted the text to fit the changed format in figure 2 and added a short paragraph about the ocean circulation in the snowball state.

Figure 3: may want to show sea ice instead/also in units of thickness/equivalent sea level, volume seems less easily interpreted here.

We now show sea-ice thickness in figure 3, as this is indeed easier to interpret. The plot of ice volume was removed, because it does not provide any additional information. We chose to show mean ice thickness instead of water equivalent, to avoid confusion about whether the value refers to a global or a hemispheric sea-level equivalent, since the plot shows the ice thickness for both hemispheres individually.

I agree with the authors that their circumpolar current is a weak point, and that this is a result of the flat topography. A mid-ocean ridge across the circumpolar opening would indeed have helped. It seems to me that the authors address this deficiency reasonably well in their analysis and discussion.

The thick snowball sea ice cover is sometimes referred to as sea glaciers, to distinguish it from the very different present-day sea ice. I don't know that this terminology is necessarily better than sea ice, admittedly.

We stick with the phrase „sea ice" when talking about the sea ice simulated in our model, because the model really only includes a formulation that was developed for the present-day sea ice. However, there are a few occasions in the manuscript where we speak about the general thick snowball sea ice cover. We adapted those formulations to „sea glacier" or similar phrases.

Lines 240-245: Interesting finding of distinct MOC cells in the freshwater and salty layers.

Figure 5: given the focus on stratification/re-stratification, it would make sense to show the temperature and density too. Perhaps another column of panels for temperature, with density contours over both temperature and salinity.

Around line 255: the density is not shown, but sloping and then vertical circumpolar salinity contours suggest that the circumpolar current is initially baroclinic and then mostly barotropic once adjusted, likely a result/artifact of the flat bottom as the authors mention. Is it? The relevance of this baroclinicity is mentioned below.

We agree that showing temperature and density is useful and adapted figure 5 accordingly. This point, regarding the sloping isolines and the baroclinic current, in combination with the clarifications later in the referees comment, is very helpful. We restructured the description of the results in figure 5 and added a few sentences.

Section 6: nice analysis of the overall warmth of climate and a useful comparison to previous studies.

Lines around 320 and 395: a very important and helpful discussion of the de-stratification timescale, and its causes, and a useful contrast with previous 1D vertical model results. This seems one of the highlights of this work. A comment on this: the important part is not the strength of the circumpolar current but its baroclinicity, given the thermal wind balance: drho/dy~du/dz. The sloping iso-halines shown at some stage of the deglaciation suggests a baroclinic current and later barotropic. The sloping lines should help the de-stratification process. Would be interesting to compare the top-to-bottom vertical shear in the circumpolar current simulated here vs in present-day and thus the implications on the sloping isolines of salinity and their contribution to the destruction of the fresh water layer.

We are thankful for this interesting and helpful comment. The referee is right that the circumpolar current is initially baroclinic and later barotropic. The strong shear in the current, together with the associated sloping isopycnals, is likely a major contributor to the fast destratification found in our model. To adequately represent this importance, we added a paragraph, including a figure showing the vertical profile of the zonal velocity in the current, to the discussion in section 7.3.

---

## Author Comment (AC2)

Response to Anonymous Referee #2

- Referees comment
- Authors response

1) The conclusion is a bit weird compared to the findings highlighted in the core of the MS. Statements presented here are speculative or not enough constrained. I encourage the authors to rewrite this section.

 lines 436-440: this statement seems to be very speculative. My main concern regards the assumption made for the post-snowball sea surface temperature. Indeed by using 15 000ppmv as a melting threshold, the authors probably underestimate the pCO$_2$ at the end of the snowball earth event, so the Earth's climate is not drastic enough (the used CO$_2$ threshold is more in agreement with the water belt solution (Abbot et al. 2011)).

Yes, the CO2 concentration we apply during the supergreenhouse climate is at the lower end of the estimates, but still a possible scenario. Furthermore, we discuss the effect of potentially higher $CO_2$ concentrations in Sec. 7.2, where we derive that even at $10^5$ ppmv moderate temperatures are conceivable near the poles. Accordingly, our statement in the conclusions says that the impeding effect of temperature was only potentially severe with very high $pCO_2$.. Hence, we do not claim that the moderate temperatures found in our study are representative for the snowball Earth aftermath in general, but that large parts of the possible range of $CO_2$ concentrations would result in climates exhibiting significantly large regions with still moderate temperatures. We rephrased this part of the conclusion to make our point more clear.

  lines 441: In addition to underestimate the pCO$_2$, this study also assumed an instantaneous ice sheets melting, so this paragraph needs to be rephrased. (we could speculate that this melting occurs on a very short period of time, 2 kyrs as defended by Hyde et al, 2000 but this behavior seems to be inconsistent with Benn et al. 2015, ice sheet-climate simulations suggesting a decreasing of the ice sheet volume with the rising of the CO$_2$ above 0,02bar).

In the mentioned paragraph, we state that the thermal expansion of seawater is small compared to the possible sea-level changes attributed to the glacio-isostatic adjustment. The melting of large continental ice sheets is of course the dominant contributor to the sea-level rise in the first several thousand years after the snowball Earth. However, including this in our model would not change the conclusion that also in the   long-term the contribution of the thermal expansion of seawater is small. Therefore, we now name the melting of the continental ice sheets in addition to the glacio-isostatic adjustments as being the dominant contributors to sea-level changes. Apart from that, we do not see a reason to reformulate this paragraph.

We do not see a conflict in the conclusions of Hyde et al. (2000) and Benn at al. (2015). Hyde et al. (2000) states that the melting of the continental ice sheet was very rapid (<2000 years) once the deglaciation started. This is what is causing the inflow of

freshwater important for the ocean stratification in the snowball Earth aftermath. The simulations of Benn et al. (2015) are about fluctuations in the continental ice sheet mass before the start of the rapid deglaciation. The reductions and oscillations they find have a much longer timescale ($>10^4$ years) and the possible freshwater input would probably have been mixed with the deeper ocean through the dynamic circulation of the snowball ocean (Ashkenazy et al., 2013, DOI: 10.1038/nature11894), before the rapid deglaciation started.

2) lines 360-374: Here the authors try to infer the time scale and environment for cap dolostone using their results about the ocean destratification. This approach is interesting but suffers a major flaw caused by the use of a uniform ocean depth (and held constant to 3500m) as a boundary condition. In my view the surface salinity simulated in the vicinity of continents (fig.7) cannot be considered as representative of coastal areas where cap dolostones were formed.

Indeed, our simplified bathymetry does not allow for a proper reconstruction of the conditions during the deposition of cap dolostones. However, we can make constrictions based on our finding of a rapid removal of the oceanic freshwater layer. As Liu et al. (2014, DOI: 10.1016/j.epsl.2014.06.039) state that there are signs of deposition in two chemically distinct fluids, we can infer that these dolostones must then have been deposited during the deglacial period, because after that, the ocean circulation would have removed any freshwater layer quickly. We reformulated the paragraph and  hope that it now gives a better explanation of the conclusions that can be drawn from our study.

- Abstract (lines 14): without an accurate bathymetry, results of this study are not robust enough to support this conclusion.

We changed the wording in this sentence, as well as in the first bullet point of the conclusions, to be in lines with the reformulated paragraph mentioned above.

- line 364 - I don't understand why the authors used Allen and Hoffman, 2005 here. This paper is explicitly focused on giant ripples recorded in cap dolostones. This paper is related to the topic but seems to be more appropriated to explore wind speeds during the sea level rise.

We agree that this citation might not be the correct reference to  say that cap dolostones where deposited in coastal areas, as we did in the text. The revised paragraph does not include this specific statement anymore and the citation is removed likewise.

3) lines 422-425: According to my understanding, the main reason of the circumpolar existence seems to be the singularity of the Marinoan paleogeography, the northern hemisphere being characterized by the absence of continents above 50° of latitudes (fig.1).

This is correct, but lines 422-425 do not discuss the reason for the existence of the circumpolar current. In the text above, around lines 405 and 419, we refer to "the location of continents during the Marinoan…" to explain the strong circumpolar current. To be more clear here, we changed this wording to "The absence of continents north of 51°N during the Marinoan…".

Minor points

- table 2. TSI → Total Solar Irradiance reduced by …

We changed the caption of table 2 accordingly.

- what's difference between aureal and austral winter?

"aureal" was a mistake, "austral" is correct. Thank you for spotting the error.

- fig. 6a "global MOC" is misleading and could be replaced by MOC at 30°S (to be consistent with the caption). fig 6b, c, d global values or zonally averaged ? (if yes you need to precise the location)

We adapted the label of figure 6a and clarified in the figure caption what properties are shown in figures 6b, c, d. Values in 6c and d are indeed global means and the transport shown in fig 6b is through a meridional section in the circumpolar current.